# ‘They Are Kids, Let Them Eat’: A Qualitative Investigation into the Parental Beliefs and Practices of Providing a Healthy Diet for Young Children among a Culturally Diverse and Deprived Population in the UK

**DOI:** 10.3390/ijerph182413087

**Published:** 2021-12-11

**Authors:** Erica Jane Cook, Faye Caroline Powell, Nasreen Ali, Catrin Pedder Penn-Jones, Bertha Ochieng, Georgina Constantinou, Gurch Randhawa

**Affiliations:** 1School of Psychology, University of Bedfordshire, Luton LU1 3JU, UK; faye.powell@beds.ac.uk; 2Institute for Health Research, University of Bedfordshire, Luton LU1 3JU, UK; nasreen.ali@beds.ac.uk (N.A.); Georgina.Constantinou@study.beds.ac.uk (G.C.); gurch.randhawa@beds.ac.uk (G.R.); 3MRC Epidemiology Unit, School of Clinical Medicine, University of Cambridge, Cambridge CB2 0QQ, UK; Catrin.Penn-Jones@mrc-epid.cam.ac.uk; 4School of Nursing and Midwifery, De Montford University, Leicester LE1 9BH, UK; bertha.ochieng@dmu.ac.uk

**Keywords:** childhood obesity, obesity prevention, healthy eating, social determinants of health, deprivation, ethnicity, beliefs

## Abstract

In the UK, ethnic minority children are at greater risk of obesity and weight-related ill health compared to the wider national population. The factors that influence the provision of a healthy diet among these populations remain less understood. An interpretive qualitative study with a phenomenological perspective comprised of 24 single sex semi-structured focus groups was conducted with 110 parents (63 mothers and 47 fathers) of young children (aged 0–5 years). The participants were recruited from deprived and ethnically diverse wards in Luton, UK and self-identified as being white British, Pakistani, Bangladeshi, black African–Caribbean or Polish. The findings highlighted a wide range of inter-relating psychological and sociocultural factors that underpin parental beliefs and practices in providing children with a healthy diet. Parents, whilst aware of the importance of providing children with a healthy diet, faced challenges such as lack of time and balancing competing responsibilities, which were clear barriers to providing children with a healthy diet. Access to and affordability of healthy food and the overexposure of cheap, convenient, and unhealthy processed foods made it increasingly difficult for parents to provide a healthy diet for their growing families. Household food practices were also found to be situated within the wider context of sociocultural and religious norms around cooking and eating, along with cultural identity and upbringing.

## 1. Introduction

Childhood obesity represents one of the most significant public health challenges of the 21st century [1] and is associated with a number of adverse health outcomes [2,3]. Children who are overweight or obese are five times more likely to stay obese into adulthood, and more likely to develop non-communicable diseases, such as diabetes and cardiovascular diseases, at a younger age [4]. The psychological burden of childhood obesity should not go unnoticed, as it is associated with increased levels of anxiety and depression, lower self-esteem, and children who are overweight are more likely to experience social problems, such as bullying and stigma [3,5,6,7]. Currently in the UK, one in three children are classified as being overweight or obese [8], with the prevalence notably higher among children from black and minority ethnic populations [9,10,11]. The economic consequence of obesity has reached crisis levels, currently costing the NHS GBP 4.2 billion a year; without urgent and radical action, this is predicted to rise to GBP 10 billion a year by 2050 [12]. The increasing number of children becoming obese, particularly those from disadvantaged backgrounds, will continue to represent an economic burden both nationally and globally. 

The social patterning of obesity, particularly among migrated populations, suggests that the environment is an important factor in explaining the increased prevalence of obesity among ethnically diverse populations [13]. Across western countries, dietary acculturation, which refers to the process where immigrants adopt the dietary practices of the ‘host’ country [14], has been strongly associated with poorer dietary intake (high in saturated fat and sugar, low in fruit and vegetables) among migrated communities compared to those who dwell within their country of origin [14,15]. For instance, both first and second generation Bangladeshi women who reside in the UK are more likely to be overweight and obese when compared to women of similar age who reside in Bangladesh [16]. Dietary acculturation is not necessarily a rapid consequence of migration, but rather a steady and progressive process that can occur and develop across generations, which is perhaps why, in particular children and in second generation immigrants, some individuals are shown to have poorer dietary health compared to more recently migrated populations [17,18]. It is instead suggested that dietary acculturation is, in fact, a dynamic process, which often begins with the introduction of the more ‘unhealthy’ accessory foods (i.e., snacks, drinks, fruits, sweets) within the household diet, with the more traditional ‘staples’ (e.g., potatoes, rice) shown to be the least receptive to change [19]. Given the negative associations between dietary acculturation and dietary health, there is a clear need to understand how dietary household food practices vary and impact minority ethnic populations.

There is also a well-documented relationship between social disadvantage and diet consumption, with a lower social status being strongly associated with an energy-dense, nutrient-poor diet with lower consumption of fruit and vegetables [20,21]. The affordability of healthy foods is consistently shown to be a core barrier among socially disadvantaged white and migrated communities, which can often favour cheap, unhealthy alternatives [20,21,22,23,24]. It is also important to note that diverse ethnic populations who reside in the UK are more likely to experience poverty when compared to the white British population [25]. In fact, current statistics suggest that in the UK, those from Pakistani, Bangladeshi and black ethnic groups are two to three times more likely to live in the 10% most income-deprived neighbourhoods compared to their white British counterparts [26]. Explanations for the links between deprivation and obesity have focused on the increased density of fast-food outlets, reduced opportunities for physical activity and the rise of social normative beliefs surrounding weight gain and obesity [27,28]. There is also some evidence suggesting that there is poor knowledge of healthy diet and weight among some minority ethnic groups [29], particularly around awareness of traditional foods that may have high levels of saturated fat and sugar [30]. The dietary attitudes, beliefs and behaviour of migrated parents are also shown to shape those of their children [31], with a common perception that their ‘traditional’ diet is healthier than the diet of their ‘host’ country [30,32,33].

Childhood obesity is multifaceted and is shown to be influenced by the complex interplay between genetic susceptibility and behaviour, primarily relating to dietary habits and physical activity [34]. However, despite a higher prevalence of obesity in children from socially disadvantaged and ethnically diverse populations [9,10,11], the influential factors among these populations remain less well understood. If interventions are to successfully target childhood obesity among minority ethnic populations, an in-depth understanding of the determinants that influence dietary beliefs and behaviours is needed [35]. Therefore, the overall aim of this study was to uncover the barriers and facilitators that help or hinder parents’ ability to provide a healthy diet to prevent overweight and obesity among their young children. We also explored, via semi-structured focus group interviews with ethnically diverse parents of young children, the influential factors that underpin parents’ beliefs and perceptions surrounding healthy foods and eating practices. 

## 2. Materials and Methods

This was a qualitative study that adopted a phenomenological perspective. This approach seeks to describe the essence of a phenomenon from the perspectives of those who have experienced it in a bid to uncover the meaning of their lived experience [36]. This study is a smaller sub-study of a larger research programme that explored parents’ beliefs and perceptions regarding healthy diet and weight for children aged 0–5 years [37,38].

### 2.1. Sampling

The participants were recruited from Luton, an ethnically diverse town located in the southeast of England, UK, with a population of just over 200,000. Luton hosts a large African–Caribbean and South Asian population [39], with 65% of the children (aged 5–15 years) who attend school in Luton being from a non-white ethnic background [40]. Luton has also seen a recent increase in citizens from Eastern Europe, most commonly Poland [39] following the 2004 enlargement of the European Union (EU) [41].

Luton experiences high rates of deprivation compared to other parts of the UK, with nine output areas in the top ten per cent most deprived in the country [42]. The sample was recruited from the five most deprived wards across Luton (as identified by the Index of Multiple Deprivation 2015) [37,43], where large sections of the ethnically diverse community reside. These wards also experience higher levels of child obesity and lower fruit and vegetable consumption compared to the averages for Luton and England [44].

A purposive stratified sampling strategy was used to recruit the focus group participants. To be eligible, the participants had to have at least one child aged 0–5 years old, reside in one of the five identified deprived wards in Luton, and self-identify as being from the white British, Pakistani, Bangladeshi, black African–Caribbean or Polish ethnic groups. The parents were included regardless of their level of English language fluency.

To recruit participants across the five wards, a systematic multi-pronged approach was adopted. Existing networks and contacts already established within the research team were used alongside community targeted advertisement and recruitment. Information and promotional materials about the study were sent via e-mail directly to all community centres and existing networks within the defined wards. Posters and other recruitment material, including leaflets, were also widely advertised throughout the wards, displayed, and handed out in local community settings, including local amenities, children’s centres, community centres and places of worship. Participants were also approached face to face via the trained and bilingual facilitators across local community groups and places of worship, with the permission of community leaders. The contact details of the research team were provided, who then contacted participants to check eligibility, discuss the study in full and address any questions.

A total of 24 single sex focus groups were conducted based on self-identified ethnicity, with a combined sample of 110 parents (63 mothers and 47 fathers). Stratifying the focus groups by age, gender and ethnicity was particularly important to facilitate and enhance engagement with participants within the diverse and hard to reach communities [45,46], whereby cultural and language barriers or gender segregation practices may have impacted on open participation [47,48,49,50]. Participants’ ages ranged between 21 and 45 years across the five defined ethnic groups: white British (*n* = 15), Pakistani (*n* = 22), Bangladeshi (*n* = 20), black African–Caribbean (*n* = 24) and Polish (*n* = 29). The number of focus groups held was driven by saturation, where consecutive focus groups revealed no additional second-level categories [51]. A breakdown of the focus groups and demographic information about the participants is provided in Table 1.

### 2.2. Data Collection

An interview guide was developed collaboratively as part of the multi-disciplinary research team to facilitate the discussions across all focus groups. The topic guide used open ended questions (Appendix A) to enable the interviewer to explore parents’ experiences of feeding their children, examine decision making in relation to current feeding practices and identify what helps or hinders them when attempting to achieve a healthy diet for their child. Probes were also used to generate further explanation where relevant.

The focus group discussions were conducted between the period of February 2014–January 2015, facilitated by experienced qualitative fieldworkers. All facilitators were bilingual and were gender matched to ensure that they could build rapport with the participants, thus allowing for more open and insightful discussions that allowed all non-English speaking participants to take part. All facilitators were recruited by the research team from the University of Bedfordshire and held a postgraduate qualification in health sciences. The facilitators were either a current PhD student at the time of the study, or an employed postgraduate researcher. A one-day training event delivered by the research team (N.A., F.C.P., E.J.C.) provided an overview of the research study, presented the research instruments to be used and provided the facilitators with an opportunity to practise the topic guide with trained researchers. Following the training event, the facilitators were asked to pilot the interview guide with two participants, not only to test the appropriateness of the questions, but also to gain experience in using the guide to build rapport and facilitate discussions. No modifications were made following this process.

The focus groups with Pakistani and Bangladeshi mothers were facilitated by bilingual facilitators proficient in Punjabi, Pahari, and Urdu. However, the participants mostly used English, with some explanations in the vernacular. All focus groups with Polish mothers were conducted in Polish and were supported by a bilingual facilitator. Black African–Caribbean and white British focus groups were all conducted in English. It is important to note that the participants did not know the facilitators prior to participation, and the focus groups were not attended by anyone other than the facilitator and the participants.

All participants were provided a participant information sheet, which clearly explained the nature and purpose of the research prior to taking part in the study. Informed consent was obtained through the completion of a consent form. The focus groups lasted approximately 90 min each and were conducted in various community locations across the five wards, including community centres and places of worship. All participants were given a £10 high street voucher upon completion of their participation in the study.

The focus groups were audiotape-recorded with the participants’ permission, translated to English by the moderator and transcribed. All facilitators were asked to check the audio recordings to confirm they were reflective of the focus groups they conducted, with all transcripts given to the participants for comment.

All qualitative data (including audio recordings) obtained from the interviews were stored anonymously and destroyed after they were confirmed as being reflective of the interview. Ethical approval was provided by the University of Bedfordshire Institute for Health Research ethics committee (IHRREC367).

### 2.3. Data Analysis

The focus groups were analysed using the framework approach (FA) [52], which is a grounded and generative analytical procedure particularly useful in identifying commonalities and differences in qualitative data [53]. The data were coded inductively and were applied manually to all narrative, which varied from small sections of data (parts of sentences) to whole paragraphs. FCP, EJC and CPJ-P then independently coded three transcripts and then discussed the labels assigned to each passage. All themes and sub themes were discussed with the authors and agreed upon after discussion. An analytic framework agreed with the research team was developed along with a brief description of all codes and was applied to all transcripts (Table 2). The final stage of analysis involved summarising and synthesising the range and diversity of coded data by refining the initial themes and categories. Abstract concepts were developed collaboratively through the identification of key dimensions of the synthesised data and making associations between the themes and concepts. NVivo v11 was used to manage the data, which allowed for ease of data retrieval and a high level of working familiarity with the data under consideration.

## 3. Results

There were six core themes that captured the barriers and facilitators to providing a healthy diet to children, which included (1) ‘knowledge, skills and capability’; (2) ‘time and financial constraints’; (3) ‘food environment barriers’; (4) ‘influence of cultural identity and upbringing’; (5) ‘family influences on eating and food practices’; and (6) ‘mealtime routine and practices’.

### 3.1. Knowledge, Beliefs and Capabilities

#### 3.1.1. Knowledge of What Constitutes a Healthy Diet

Parents, irrespective of ethnicity, had a good general understanding of what constitutes a healthy diet. Perceptions centred on a balanced diet with a high consumption of fruit and vegetables, with a strong importance held among many of the ethnic groups on natural, organic, and fresh food. Knowledge regarding the role of other major food groups, such as protein, carbohydrates, fats, sugar, and salt, was less evident, with Polish and white British parents shown to be the most aware. Discrepant beliefs and knowledge surrounding the role of sugar and salt were also evident, which revealed both demographic and cultural differences. For example, whilst most parents felt that sugar should be avoided, Pakistani and Bangladeshi parents felt that sugar and salt were an essential aspect of enhancing flavour: ‘with Asian food, the salt does play a big part. If you haven’t got enough salt, you don’t get all the spices’ taste, that’s what it comes down to’ (Pakistani father, aged 31–45).

#### 3.1.2. Benefits and Consequences of a Healthy Diet

Parents also had a good awareness of the benefits or consequences of maintaining a healthy or unhealthy diet for them and their children. Black African and Polish fathers felt that a healthy diet is also directly linked to how we feel, with a healthy diet used to promote good emotional and physical health, and therefore felt that providing a child with a healthy diet was extremely important.

#### 3.1.3. Capability to Prepare and Provide a Healthy Diet

Whilst mothers disclosed that they had good knowledge, confidence, and skills to prepare and cook healthy and nutritious meals for their children, this was less evident among fathers. Many fathers were very open about not being involved in food preparation or cooking, and lacking experience and practical skills around cooking and food preparation: ‘yeah no, like I said, I don’t do any of the kitchen work, I don’t do any kitchen work. [partner/name removed] does it all… to be honest I can’t cook. Well, I can make biscuits, but that’s it so yeah, she does it all’ (white British fathers, 21–30). Nonetheless, black, and white British fathers were receptive to being more involved in the cooking of the household: ‘stick me on a cooking lesson man innit, nah, if I could cook, I’d probably cook. I like trying to cook’ (white British father, 21–30). South Asian fathers, in contrast, as discussed below, felt that any support should be tailored towards the mothers, as within their community the woman is solely responsible for all aspects relating to the household diet.

Participant 1: ‘Because this is where the problem lies, I think, with the Asian mothers. They have so many other responsibilities and if they were to get taught a course, or they could go onto a course where they could possibly be taught how to try to prepare their child’s food and the healthier version, I think that would help’.

Participant 2: ‘Yes, that’s where the focus should transfer because most of our families, it’s women who are dealing with all the cooking, all the stuff, the feeding gets done, all those sorts of things’ (Pakistani fathers, aged 31–45).

#### 3.1.4. Practical Skills and Strategies to Overcome Time and Financial Barriers

Strategies to overcome the lack of time and competing work responsibilities were widely discussed. Many Polish and white British mothers considered the use of slow cooking as a useful strategy and would prepare foods in the morning, which they would then leave to cook on a low heat whilst at work: ‘I just put everything in the pot, set it on low for eight hours and then it is done. It just takes out all the stress and I know my family will have a healthy meal when they get home’ (white British mother, aged 31–45). Meal planning was also considered an important strategy, which could be adapted to the family’s schedule: ‘I write up a very exact list of what we will be eating for dinner every day’ (Polish mother, aged 31–45). Workshops to support parents with food preparation skills to reduce time, including batch cooking and freezing portions, were discussed among some parents, alongside the use of menu cards with ingredients and recipes to support parents with planning (identifying foods required) and preparing meals.

### 3.2. Time and Financial Constraints to Healthy Eating

#### 3.2.1. Lack of Time to Provide a Healthy Diet

The mother was mostly viewed as being responsible for the household diet and, consequently, all aspects relating to the child’s diet. The only contrast to this was found among Polish parents, where fathers revealed that the household diet was rather a joint commitment between both partners who instead worked towards a ‘*collective positive goal together*’. Many women across the focus groups (white British, Polish, African–Caribbean) worked, so whilst many mothers acknowledged that cooking fresh food from scratch was the healthiest option, the lack of time made many of these women feel like it was simply unachievable. As one mother stated: ‘life confronts all knowledge we are given and puts it to the test’ (white British mother, aged 31–45). Many of these working mothers felt that whilst they had the best of intentions, the reality of exhaustion from balancing work and childcare made preparing and cooking fresh food for the family feel simply impossible: ‘I’m so tired after work, I just give them crisps. I am just not strong enough to cook’ (black mother, aged 31–45). African mothers also discussed the challenges of cooking traditional cultural dishes and the added time it takes in food preparation, representing an additional significant barrier to preparing healthy foods for their children: ‘… time it takes in the preparation of yam and plantain, which need to be grinded. Green ones you need to grind, but do people really have time to be grinding?’ (black African mother, aged 21–30).

#### 3.2.2. Affordability and Accessibility of Providing a Healthy Diet

The affordability of food was also widely discussed as a potential barrier to healthy food consumption, with a consensus, irrespective of ethnicity and gender, that meat, fresh fruit and vegetables and organic foods were more expensive than processed and unhealthy alternatives: ‘If we went organic, for example, that would be the best way, but it costs money. [participants agree] So when we talk about the community, not many of them can afford that particular way’ (Pakistani father, aged 21–30). However, for some (Pakistani) parents, the cost was viewed as an excuse: ‘we can spend money on clothes, how many dresses do we have? And we say that we have limited money to spend on food. These are excuses, I think… I wouldn’t care about the cost; I would just want my baby to eat’ (Pakistani mother, aged 21–30). Pakistani fathers argued that cost was not necessarily a barrier, but rather it was about parents resetting their priorities: ‘in a sense, we don’t look at the prices, we don’t look at the cost, we don’t look at how far we have to go to get that item. We will go out and do it because we don’t have a habit of going out everywhere, or maybe every other day, going out in the clubs or a pub where you can have a lunch or something like that, we stay inside the house. We take the food into whatever places as well. The cost is minimal’ (Pakistani father, 31–45). Black African–Caribbean parents further discussed the difficulties surrounding the affordability and accessibility of important cultural foods, including fruits (e.g., plantain, green bananas), vegetables (e.g., okra, yam) and other traditional speciality foods (e.g., maize meal, èbà, àmàlà and swaloo). Many parents would have to use online delivery methods to source these foods; however, the high cost often meant that parents had to rely on unhealthier ‘westernised’ food substitutes.

The Healthy Start voucher scheme was widely discussed across the Pakistani, Bangladeshi and white British focus groups. This national scheme provides monetary vouchers to eligible parents (for children up to the age of four years) to spend every week on milk, fresh and frozen fruit and vegetables and vitamin supplements. Parents who accessed this scheme felt that it provided a useful financial assistance; however, awareness among some parents was limited. White British fathers were less aware of this scheme, including the eligibility criteria to enrol onto the scheme and/or that it included fruit and vegetables (many fathers thought it only included milk). Pakistani and Bangladeshi mothers also revealed that they could not use the vouchers in their local shops, so whilst they were entitled to use the vouchers, they were often left unredeemed. There was also consensus across all of the focus groups that this scheme should be extended to older children, particularly for low-income families. 

### 3.3. Food Environment Barriers

#### 3.3.1. Cost of Unhealthy Food

Many parents, particularly white British fathers and black African mothers felt that fast food was a cheaper alternative to buying fresh food and cooking at home: ‘If you buy meat and potato, it costs £10. For £5, I can buy a bucket (of chicken and chips)’ (black African mother, aged 30–45).

#### 3.3.2. Exposure of Unhealthy Food

Takeaways were not only seen as a cheaper option, but also a more accessible and convenient one. Many parents felt that there were too many takeaway restaurants and convenience stores in the local area, and that whilst it may not influence parental choices for their children now, they feared that it may affect the decisions their children will make when they are older: ‘it’s frightening, ‘cause you realise that when they start working, it’s a whole different ball game… their entire money will be spent on fast food. They have no interest in, like, cooking, like they say, “oh there’s a restaurant—let’s go to the restaurant”’ (black Caribbean father, aged 31–45). Polish fathers also discussed the limited healthy options when taking children out: ‘So, let me give you an example. Let’s say someone is organising a kids’ party in some omnipresent place, some fun factory or something, and there someone has created a kids’ menu and there are five items to choose from, whereby all five positions listed on there is basically poison that you are feeding your children’ (Polish father, aged 21–30).

There was an over-reliance, particularly among younger parents, to rely on takeaways, to cook frozen and processed food and/or to eat out more frequently. As stated by one young white British father: ‘I’m just lazy. I can’t be bothered to cook, so I just order food. I’m not even going to lie, like, or I’ll put a pizza in the oven’ (aged 21–30). Whilst it was accepted that this may be a potential contributory factor for obesity levels in children, the convenience and availability of convenience foods appeared to have made this a social norm. Recommendations among parents centred on environmental adaptations, with a need to discourage and limit fast food. Polish parents had particularly strong views, suggesting that fast food should be banned with laws or sanctions placed on parents to restrict the types of food they purchase: ‘but they have an addiction, just like someone had an addiction to cigarettes, and that costs the NHS—cancer and that, and obese people cost the NHS—and in association with this, we should ban all forms of fast food, and who allowed it to be sold for a pound?!’ (Polish mother, aged 31–45).

### 3.4. Influence of Cultural Identity and Upbringing

All parents, irrespective of ethnicity and gender, discussed how their childhood experiences and upbringing and cultural identity influenced their approach to child feeding, which both facilitated and challenged their ability to provide a healthy diet for their young children.

#### 3.4.1. Past Childhood Experiences

Past childhood experiences were a particularly important influence on current family food practices among South Asian and black parents: ‘yeah, but mostly he likes our traditional food, because that’s what he was brought up on—so really, it is the bringing up, yeah’ (black mothers, aged 31–45).

Whilst some parents revealed their childhood experiences shaped the way they do things with their children, white British and Polish mothers were more critical of their parents’ approach and, where relevant, tried to adapt their feeding practices or their child’s dietary intake in areas where they felt things could be improved. White mothers were particularly reflective about feeling pressured to eat when they were younger, so there were many discussions around trying to provide their children more autonomy around satiety: ‘I just didn’t want to have a situation where Violet was forced to eat beyond her point of hunger [Participant 2: yep!], beyond, you know, she’s full or she feels she’s had enough. I want her to feel to say I’ve had enough’ (Participant 2). A few Polish mothers revealed that they did not eat as a family, so they have incorporated this as an important family ritual: ‘In my house, it was the case that we very rarely ate dinner together [as a family]. My parents were constantly at work, and I had the house keys and so did my brother, so we would return home [and eat] at various times. So, then it is the case that in my house, we will eat dinner together, or at least we will try to, and that had an impact on this’ (Polish mother, aged 31–45).

#### 3.4.2. Quality of Healthy Food/s

Across all black, South Asian and Polish focus groups, parents stated that they wanted to bring their children up on a culturally traditional diet to reflect their childhood. The parents wanted their children to learn and engage with their ethnic heritage, as this formed an important part of their cultural identity. However, there were some challenges that parents discussed around adapting their cultural diet within the UK. Many parents felt that the quality of food in the UK, particularly around fruit and vegetables, was not comparable to their ‘home’ country. Polish and African parents felt food in the UK was mostly pre-packaged and had a different taste, which has led to a change in taste buds among their children: ‘In the West Indies, you would have passed, like, let’s say 20 different fruit trees, or 10 different fruit trees; it could be wild berries, it could be guava, it could be—all these things are high in different nutrients and everything, so within that time we would have eaten, you know what I mean, we just grab it and eat it naturally. But kids’ taste buds here, you give them certain fruits and they don’t like the taste of it, you know what I mean, because their taste buds haven’t—I don’t know, I can’t say developed, but they like something that is sweet because the food here is being processed to taste a certain way. A banana, in this country, some bananas are processed to taste different’ (black African father, aged 21–30).

#### 3.4.3. Adaptation to a ‘Western’ Diet

Polish parents discussed how they modified their family’s diet to incorporate English foods, with this being viewed important to allow their child to adapt to their environment. Black African –Caribbean and South Asian parents were more likely to adopt their traditional cultural diet. However, the compatibility of their traditional diet within the UK was widely debated.


*It is a bit different because the climate there [Pakistan] is different. It’s hot, you’re moving up and down, you’re sweating it out. Here, it is not the same. So, they are trying to give us food that you can take there and probably use that, burn it off in that country, but here it is not the same. They give the same food that we cannot burn off. That is why we’ve have got so many problems.*
(Pakistani Father, aged 21–30)

Further, cultural differences emerged among the participants on how their dietary practices compared to those in the UK. The discussions mostly focused on portion size, where in South Asian and African-Caribbean cultures, people would eat more. One Pakistani father (aged 31–45) stated: ‘you said earlier, when brother was saying about limit, what limit you should give, you shouldn’t be using that word limit. For Asians, there is no limit! No matter which household you go to, they say [in Urdu] “they are kids, let them eat”, so there is no limit… English people, they would be very blunt. They would say to your face, no, and then you will not, that no means no, for Asians, no means [in Urdu] give them more’. There were discussions, most notably among younger South Asian parents, that they did not need as much food as they had been educated by their elders to eat. Having this awareness enabled the parents to be more mindful around portion sizes, particularly when feeding their children.

#### 3.4.4. Compatibility of Cultural Diet with Health Promotion Campaigns

Another potential barrier discussed by parents surrounded health promotion messages. There was a lack of awareness among black and South Asian parents regarding the most important health concerns that impacted their communities: ‘culturally, there is a difference with the priorities, and there aren’t always people around to teach other cultures about things, like high blood pressure or diabetes or eating disorders. Some people just feed their kids with what they know. I am kind of guilty for that until recently. It isn’t a form of neglect, just following tradition’ (Black father, aged 31–45). South Asian fathers also felt that promotional campaigns lacked cultural relevance, for example, the ‘5 a Day’ promotion: ‘It’s really hard to please the kids with that food, the so called 5 a Day. It’s very, very hard because we’re not used to it. Perhaps other people know how to cook it that way, so the kids have got the taste to it, but our kids, as an Asian—not really’ (Pakistani father, aged 31–45). The parents, therefore, felt that these health messages could be more culturally tailored to their community, with more guidance and practical advice on how these recommendations can be adapted to their cultural diet without jeopardising the taste. There was, nonetheless, an acknowledgement among these fathers that adapting the cultural diet was not something that could be changed overnight, and instead was viewed as a slow natural process that would occur across generations.


*I don’t think the Asian community can change it. I think the only way it will happen is if it fades out itself, over generations it fades out. I think it’s to do with culture, the culture we’re getting from is Pakistan and Kashmir and obviously our first generation that came, some of us second generation, I think slowly it will fade out, it’s time. I don’t see how you could change the ideas of a 60-year-old and try to convince them that what they’re doing is wrong, especially culturally. Again, going back to the culture, it’s not considered as good etiquette, either.*
(Pakistani father, aged 21–35)

### 3.5. Family Influences on Eating and Food Practices

#### 3.5.1. Influence of Extended Family on Eating and Food Practices

All parents, irrespective of ethnicity and gender, discussed how their family members influenced their approach to child feeding. The parents felt that family both facilitated and challenged their ability to provide a healthy diet for their young children.

Family members were mostly viewed as a positive influence. Many mothers (particularly black African–Caribbean, South Asian and Polish) valued the experience held by their parents and the support and knowledge they could offer them: ‘my biggest influence would also be my mum, my mum, definitely, if anything, my mum. Because she is experienced, and she raised, like, us lot, my brothers, sisters and me, she’s got the knowledge. She will be the biggest influence for me’ (Bangladeshi mother, aged 21–30). This was further supported by fathers, who particularly valued the role of the grandparents (particularly their mother), with black fathers viewing them as an important ‘healthy’ role model with whom they share their cultural background: ‘we don’t really take mainstream information, as our grandparents are there to guide us should any issues arise. We were bought up healthy, we follow the regime our culture and forefathers use’ (black father, age unknown).

However, some challenges were noted, particularly surrounding discrepant views of grandparents. For example, the Pakistani and Bangladeshi parents revealed they had differing views between them and their elders towards food choices for their child/ren. One Pakistani father discussed how older generations would have had different priorities, which impacted on their views towards healthy food: ‘If any of our parents are here in this country, they’ve come over 20/30/40 years ago and it wasn’t about looking into “My child needs to get his 5 a day”, it was “I need to be able to feed my child”, so I think back then that was the main priority, feeding my children, not deciding on “I need to make sure I can feed my child five different things”, and it stems from that, and obviously our parents then, if you try to say “We want to try to give our child XYZ” or we want to try and give them some healthy options, your father will say “what’s wrong with the dhal and roti that we’ve been eating for the last 30/40 years? Nothing happened to us” (Pakistani father, aged 31–45).

Many parents also felt that grandparents would undermine them by giving their child/ren ‘treats’, regardless of what they are allowed at home, which would cause issues: ‘When she goes to grandparents’ house, [in Urdu] “it’s okay, let me give them, you want more, sweetheart?” They don’t stick to the guidance, e.g., five a day’ (Pakistani father, aged 31–45). Some (Bangladeshi) fathers also felt they can be opinionated, and this presented a challenge, particularly when they disagreed.

Polish parents, whilst they valued the grandparents’ involvement, did discuss the impact of geographical distance on their relationship whilst living in the UK. As first-generation immigrants, the Polish parents felt that they had more freedom regarding the food choices they could provide for their families. Whilst grandparents would be involved with the children’s diet through phone conversations, this was not viewed to have the same effect. As one Polish father (aged 31–45) stated: ‘The in-laws are the biggest treasure [of knowledge] and it cannot be avoided, especially amongst Poles, where I believe in-laws, i.e., the grandmothers, have the most to say. Eh, they are the mothers who raised children. Even though we are so far away, on Skype, those mothers/grandmothers would gather around and would give advice via Skype, and they would ask what’s with the child or what he been eating, eh, what do we feed the child? And, of course, when it comes to a home visit [by the in-laws], well, then unfortunately, we have no say. Just my mother and mother-in-law take control of the child’.

#### 3.5.2. Taste Preferences of the Child/ren

Discussions across all ethnic groups focused on the role of the children’s taste preferences and the impact this can have when trying to provide a healthy diet. Many parents revealed that their child would refuse healthy foods, particularly fruit and vegetables, in favour of more ‘palatable’ and unhealthy options. Below is an extract from black African fathers (aged 21–30) discussing the reasons why children do not eat five portions of fruit and vegetables a day.

Participant 1: ‘Not having five a day, fruit doesn’t taste good.’

Participant 2: ‘But do you think your children have five fruits a day?’

All participants: ‘No, no.’

Facilitator: ‘And why do you think that is?’

Participant 1: ‘I would say like most of the kids, they’re not into the fruits.’

Participant 3: ‘The fruits don’t even taste good to them anyway, so I think that’s off-putting as well, that sort of taste.’

The parents felt that children’s taste preferences were inherent, unique to the individual child and were not resistant to change. Therefore, most parents (black, South Asian and white British) managed their child’s taste preferences by accepting their dislikes and offering foods that they knew they liked: ‘I’ll try and get him to eat vegetables and that, but if he don’t like them then there’s not really much I can do about that… you can’t force a kid, so you just end up giving them what they like’ (Bangladeshi mother, aged 21–30). In these situations, parents appeared to be more concerned that their child was full and perhaps less concerned on the types of foods they were eating. For example, a Bangladeshi father discussed his child who often refuses food at school: ‘kids don’t like the bland food without salt in at school, so they don’t eat and come home and eat junk and be obese. I’d rather them eat junk food than nothing at all’ (Bangladeshi father, 31–45). Overall, there appeared to be a shift towards giving children more control and autonomy over their food choices. This was less evident among Polish parents, who, whilst acknowledging that their children had certain preferences towards more unhealthy snacks, said these were only given if they were eating their meals: ‘it’s the same situation with us, but we just do not give it to them. We just tell them not to eat it, and to eat the right meals they are supposed to on a regular basis’ (Polish father, aged 31–45).

### 3.6. Mealtime Routines and Practices

#### 3.6.1. Role Modelling

Some parents discussed the positive impact of different feeding practices used by them to promote healthy eating in their children. The parents placed particular emphasis on being a role model for their child. For example, the Polish parents discussed the importance of parents as role models to children: ‘we are examples, and I think that I am a healthy person and so, well, I feel that I have an advantage due to this [being healthy] and that the child should have the same basis [of an upbringing] as I did’ (Polish father, 21–45). The Polish parents also emphasised the importance of changing the whole family’s diet to encourage children to eat a healthy diet. Pakistani mothers felt that siblings were important influences on children, whereby they would want to copy what others are eating at the dinner table: ‘when my baby also sees that his older sister and brother are eating some different food, he would leave his food aside and wouldn’t even touch it. He would like to taste what they are eating’ (Pakistani mother, aged 31–45). Black fathers also discussed the importance of talking to their children about foods that the adults are eating and explaining the benefits: ‘We try to explain the benefits of everything that we give to them, so they understand. We let them see us eating certain vegetables’ (black father, 21–30).

#### 3.6.2. Structure and Routine around Mealtimes

Having a clear structure and routine around family mealtimes was also viewed as an important facilitator for improving healthy eating habits. As one Bangladeshi father stated: ‘I think that’s a personal problem because, I’ll be honest, I don’t have much of routine myself and I think that it goes two ways as well to the child, ‘cause he seems to snack just ‘cause he sees me snacking, or someone comes round, he wants his sweets’. However, South Asian parents felt that mealtimes, like taste preferences, should be led by the child: ‘there are general things that you can follow, general advice, but what your child likes, what timetable they have for eating and that kind of stuff, you can’t control that. It’s child-centred, whatever they would like’ (Pakistani father, aged 31–45).

White British fathers discussed the challenges of ‘co-parenting’ and the impact this had on providing a child with a consistent routine: ‘it’s different, obviously, like, as soon as you split up with your partner because they do it different to you than what you do, so yeah, and if one day you’re feeding him one sort of stuff that’s good, like, she could be doing something completely different’ (white British father, 21–35). The change of environment when a child was living across two households was viewed as a barrier that fathers had to adapt to: ‘It [child attending father’s house] changes the routine, I don’t know when she’s at her mums what she eats, but with me, she was refusing to eat anything healthy, but she’s out of that now she’s been following my routine and rules’ (white British father, aged 21–35).

## 4. Discussion

This study provides a current qualitative exploration of parents’ experiences of providing their children with a healthy or unhealthy diet in a culturally diverse and deprived community. There were six core themes that captured the barriers and facilitators to providing children with a healthy diet, which included (1) ‘knowledge, skills and capability’; (2) ‘time and financial constraints’; (3) ‘food environment barriers’; (4) ‘influence of cultural identity and upbringing’; (5) ‘family influences on eating and food practices’; and (6) ‘mealtime routine and practices’, which provide a valuable insight into the factors that inform and underpin parental beliefs and practices.

Whilst the parents were aware of the importance of providing their children with a healthy diet, and were motivated to so, the perceptions surrounding lack of time and exhaustion through many competing responsibilities made many mothers feel that providing a healthy diet for their family was simply unachievable. Many mothers revealed they worked long hours, sometimes doing shift work, which they had to balance with childcare alongside being solely responsible for managing the dietary needs of the household. This finding supports broader international evidence that highlights that these barriers among migrated populations have been strongly related to a range of dietary lifestyle behaviours, including poor diet and lower levels of physical activity [54,55,56,57]. Strategies and practical support for parents in relation to meal planning and preparation were viewed as extremely welcome, alongside the provision of recipes for meals that are culturally nutritious for the family, yet quick and convenient for parents to make.

The findings also revealed that many of the parents we interviewed may be facing food insecurity due to the difficulties of accessing nutritionally healthy and culturally appropriate foods to meet their family’s dietary requirements. Affordability was cited as the most common barrier, particularly of fresh fruit, vegetables and organic foods. The accessibility and cost of cultural specialty foods was also a significant challenge, particularly among black African–Caribbean families. This challenge becomes increasingly problematic when these families are exposed to unhealthy yet cheaper foods in their immediate environment. The overexposure of cheap, high calorie and low nutrient energy-dense foods in their local community was widely discussed, with takeaway establishments becoming a more convenient, and socially acceptable, option for many of the families we interviewed. These challenges combined make it increasingly difficult for parents to provide their growing families with a healthy diet. Evidence-based nutritional strategies that are community driven would be well placed to reduce the risk factors associated with food insecurity and promote sustainable approaches for health food access, particularly among the most vulnerable [57].

Household food practices were found to be situated within the wider context of sociocultural and religious norms around cooking and eating. The importance of raising children on a traditional cultural diet was particularly important among black African–Caribbean and South Asian parents. However, discussions regarding how traditional cultural dietary practices often conflicted with those in the UK revealed that routines, composition of diet and approaches to diet were widely recognised. The findings also uncovered, as supported by previous research, that some generational differences—most notably among black African–Caribbean and South Asian parents—were evident, whereby some parents felt pressured by their older generations to adhere to traditional food practices [21,58]. However, interestingly, less pressure was felt among Polish parents who were living in the UK as first-generation immigrants, where acculturation to English food was viewed as important to allow their child to adapt to their host environment. Whilst less is known regarding how dietary practices vary among Polish migrants, it has been suggested that younger migrants may be more flexible and more tolerant of changes in dietary habits [23].

Children’s food preferences, particularly the refusal of fruit and vegetables in favour of more ‘palatable’ and unhealthier options, was a notable barrier discussed across all of the focus groups, a finding that has been identified across other migrated populations globally [14,59,60,61]. Children have been shown to have a genetic predisposition and evolutionary derived preference for sweet foods and a dislike of bitter foods [62], which can vary by age. Food preferences and food choices among populations are further linked to attitudinal, social and economic variables, such as income [63]. The most common strategy disclosed among the black African–Caribbean, Pakistani and Bangladeshi parents in addressing this focused on offering their infant an alternative option. This supports literature that has highlighted the conflict that many migrated mothers may feel between their feeding choice and the child’s food preference, which is often resolved through preparing foods their children like [61]. However, more recent evidence suggests that akin to the approach that the Polish parents used, parent-led exposure and incentives to encourage children to taste unfamiliar foods has been shown to be a more promising strategy for promoting liking of previously rejected foods in children [64,65]. Interventions that seek to provide tailored support and guidance for parents to manage taste preferences and food rejection may, therefore, equip parents with the skills to enable their children to make healthier food choices.

The importance of increasing the representation of socially disadvantaged communities in research is pivotal to ensure that targeted public health interventions reflect the wider ethnically diverse population. This study, through the employment of a tailored and flexible recruitment strategy, enabled us to achieve excellent participation with commonly underrepresented and hard-to-reach groups, and enabled a richer understanding of views and experiences reflective of the wider community, a common limitation of previous research [66,67]. This study has also complemented the existing research base through the inclusion of Polish perspectives; despite increasing migration and Poland being most common non-UK country of birth [68], there still remains a limited understanding of the barriers that these communities face.

However, it should be noted that there were challenges in recruiting black African fathers and black Caribbean mothers, thus any interpretation and use of these findings should acknowledge that these are not exhaustive in their inclusion of all ethnic groups in Luton. The research design, through the inclusion of the critical dimensions of age, sex and ethnic variation and the use of the framework method to analyse the data, enabled us to identify and reveal useful and important patterns of similarities and differences where they exist. This has provided a useful insight into the patterns that exist whilst highlighting the need for avenues of further exploration. A more inductive approach could have uncovered a different interpretation of the focus groups, as is typical in qualitative research. We believe, however, that as this work aimed to inform practical interventions, this approach ensured findings were grounded in the data whilst answering important questions commissioners may have for intervening in these communities.

## 5. Conclusions

This research aimed to explore parents’ experiences of providing their young children with a healthy or unhealthy diet in a culturally diverse and deprived community. The findings highlighted a wide range of inter-relating psychological and sociocultural factors that inform and underpin parental beliefs and practices related to the provision of a healthy diet. Overall, while parents were aware of the importance of a healthy diet, challenges, notably lack of time and balancing competing responsibilities, were clear barriers to providing a healthy diet for their young children. Access to and affordability of healthy food, alongside the overexposure of cheap, convenient and unhealthy processed foods, also made it difficult for parents to provide their growing families with a healthy diet.

Culturally tailored strategies and practical support for parents, particularly surrounding meal planning and preparation, were welcomed, alongside more support on how to prepare culturally nutritious foods acceptable to their families that remain quick and convenient for the parents. Environmental and community driven strategies would also be well placed to reduce the risk factors associated with food insecurity and promote sustainable approaches for healthy food access, particularly among the most vulnerable in society. The role of the child in shaping food choices should also be noted, with more adequate support and advice for parents in managing taste preferences and food refusal.

## Figures and Tables

**Table 1 ijerph-18-13087-t001:** Sample breakdown for all focus groups by ethnic group.

Ethnicity	Focus Groups (*n*)	Total
Pakistani	Male (2) Female (2)	22
Bangladeshi	Male (2) Female (2)	20
Black African	Male (2) Female (2)	15
Black Caribbean	Male (2)	9
Polish	Female (3) Male (4)	29
White British	Female (2) Male (1)	15
**Total**	24	110

**Table 2 ijerph-18-13087-t002:** Analytic coding framework with definitions.

Code	Description
Knowledge, skills, and capabilities	
Knowledge of what constitutes a healthy diet	Awareness of a healthy diet, gaps in knowledge
Benefits and consequences of a healthy diet	Perceived benefits, consequences of having a healthy diet, importance of a healthy diet, community priorities
Capability to prepare and provide a healthy diet	Beliefs around capabilities, skills to prepare and cook healthy foods
Practical skills and strategies	Strategies to overcome time management, how to cook with healthy foods, prepare cheap and nutritious meals, food preparation
Time and financial constraints	
Lack of time to provide a healthy diet	Perceived time (or lack of) to prepare and cook healthy foods/meals, competing priorities (e.g., work, childcare, caring)
Affordability and accessibility of providing a healthy diet	Cost of healthy foods, financial support with purchasing healthy foods, cost of culturally sourced foods, voucher/financial assistance to purchase healthy foods
Accessibility of culturally healthy foods	Accessibility of traditional specialty foods
Food environment barriers	
Cost of unhealthy food	Perceptions of cost of processed/unhealthy food in comparison to healthier foods,
Exposure of unhealthy foods	Availability of takeaways, fast food establishments in local environment, exposure to unhealthy food when out, perceived convenience of purchasing foods vs. to buy and prepare healthy foods, availability of healthy foods when out
Influence of cultural identity and upbringing	
Past childhood experiences	Perceptions (positive and negative) towards eating practices as child, challenging previous experiences, and making adaptations
Quality of healthy foods	Perceived quality of fresh foods (fruits, vegetables) compared to ‘native’, micro-nutrient density of foods, taste differences and adaptations,
Adaptation to a ‘western’ diet	Modifications of traditional diet, inclusion of ‘western’ foods into diet, adaptations of dietary practices (e.g., portion size)
Compatibility of cultural diet with health promotion campaigns	Cultural relevance/challenges of adapting traditional diet to adhere to health promotion campaigns/promotion
Family influences on eating and food practices	
Influence of extended family on household food practices	Positive and negative influences, respect of knowledge and experience, cultural role models, discrepant views
Taste preferences of child	Taste preferences, children’s desire (or not) to eat healthy/unhealthy foods, reasons for taste preferences, impact of taste preferences inc. child’s refusal of foods, parental strategies to overcome refusal of food
Mealtime routines and practices	
Role modelling	Parental influences on children’s diet, role of sibling/s, role modelling healthy eating practices, talking about healthy foods
Structure and routine around mealtimes	Set routine of mealtimes, household rules around snacking between meals, family led versus child led routines, challenges around co-parenting

## Data Availability

The datasets generated and analysed during the current study are not publicly available due to confidentiality, but are available from the corresponding author on reasonable request.

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
