# Peer review of "‘They Are Kids, Let Them Eat’: A Qualitative Investigation into the Parental Beliefs and Practices of Providing a Healthy Diet for Young Children among a Culturally Diverse and Deprived Population in the UK"

_ijerph, 2021, doi:10.3390/ijerph182413087_

Round 1

Reviewer 1 Report

The goal of this paper is to investigate parental beliefs and practices related to providing healthy diets to infants and pre-school children among an ethnically diverse, low-income population in the United Kingdom. The topic of this paper is of importance given the international burden of obesity and other non-communicable diseases, and ongoing preventive efforts targeting different populations at all life stages. While this paper and topic would be of interest to the readership of IJERPH, there are some concerns/questions that should be addressed.

First, while I appreciate that the authors were trying to capture a broad range of age, sex and ethnic groups, there were many different “unique” groups. To understand perceptions and phenomena of interest in qualitative research the concept of saturation usually comes into play. It seems like each of these groups may possibly have different experiences, perceptions etc. which would make it hard to fully capture just with one focus group. Furthermore, it seems like the authors, in their results, are comparing across these groups. This is not described in the analysis and again seems that comparing only one group vs. another may be inadequate. The authors should also provide further justification as to why the group stratification- of course there were language/cultural reasons to do this but what other differences were expected. They mention Pakistani, Bangladeshi, Black in the background but not the other groups.

Second, more information regarding the facilitators, their training/credentials, and participant knowledge of the interviewer would add more information to the reader about personal characteristics of the research team and reflexibility.

Third, it was unclear what methodological orientation was utilized to underpin the study (grounded theory, discourse analysis, etc?) and if the principle of saturation was used as well. Explicitly stating these processes would provide the reader with more insights related to the study design.

Fourth, while I understand that the study is very specific to UK and these ethnic groups- there is a broader literature with regards to immigrants, change in eating, food, parenting etc. that could be included in the discussion to provide additional comparison/context.

For example:

Immigrating to the US: what Brazilian, Latin American and Haitian women have to say about changes to their lifestyle that may be associated with obesity.

Tovar A, Must A, Metayer N, Gute DM, Pirie A, Hyatt RR, Economos CD.

J Immigr Minor Health. 2013 Apr;15(2):357-64. doi: 10.1007/s10903-012-9665-8.

Feeding decision-making among first generation Latinas living in non-metropolitan and small metro areas.

Pineros-Leano M, Tabb K, Liechty J, Castañeda Y, Williams M.

PLoS One. 2019 Mar 18;14(3):e0213442. doi: 10.1371/journal.pone.0213442. eCollection 2019.

Factors associated with feeding practices of black immigrant mothers of African and Caribbean origin living in Ottawa, Canada.

Kengneson CC, Blanchet R, Sanou D, Batal M, Giroux I.

Appetite. 2021 Dec 1;167:105641. doi: 10.1016/j.appet.2021.105641. Epub 2021 Aug 10.

How a Racially/Ethnically Diverse and Immigrant Sample Qualitatively Describes the Role of Traditional and Non-traditional Foods in Feeding Their Children.

Trofholz A, Richardson K, Mohamed N, Vang C, Berge JM.

J Immigr Minor Health. 2020 Dec;22(6):1155-1162. doi: 10.1007/s10903-020-00999-3.

PMID: 32219660

We recommend that the authors consider referencing the COREQ checklist as this may add rigor to their process reporting: http://cdn.elsevier.com/promis_misc/ISSM_COREQ_Checklist.pdf.

More detailed comments

Abstract: the first sentence (lines 16-17) could be broken up into two sentences to increase clarity for the reader.

Materials and methods:

  • Data collection:
    • Line 140: was this guide pilot tested? If so, need to delineate the process of pilot-testing.
    • Lines 155-157: this sentence is a little unclear. It seems like the focus groups were facilitated in English, although the facilitators were proficient in Punjabi, Pahari and Urdu; would break the sentences or reword for clarification
    • Line 170-171: how were the audio recordings confirmed as being reflective of the interview? Did the facilitators give confirmation or were the researchers?
    • Please provide justification for same sex groups
  • Discussion: line 585 – unclear if it’s a title?

Author Response

R1 C1: First, while I appreciate that the authors were trying to capture a broad range of age, sex and ethnic groups, there were many different “unique” groups. To understand perceptions and phenomena of interest in qualitative research the concept of saturation usually comes into play. It seems like each of these groups may possibly have different experiences, perceptions etc. which would make it hard to fully capture just with one focus group. Furthermore, it seems like the authors, in their results, are comparing across these groups. This is not described in the analysis and again seems that comparing only one group vs. another may be inadequate.

Response: We appreciate that the reason for stratifying and trying to capture different groups is an important aspect of this study and we regret that this was not made clear.  We have now added this section to the methods to hopefully help provide the reader a justification of this important decision ‘A total of 24 single sex focus groups were conducted based on self-identified ethnicity, with a combined sample of 110 parents (63 mothers and 47 fathers).  Stratifying focus groups by age, gender and ethnicity was particularly important to facilitate and enhance engagement with participants within the diverse and hard to reach communities [44, 45] whereby cultural and language barriers or gender segregation practices may have impacted on open participation [43, 46-48]’. We should also note that the principle of saturation was used which has now also been added: ‘The number of focus groups held was driven by saturation, where consecutive focus groups revealed no additional second-level categories [39]

R1 C2: The authors should also provide further justification as to why the group stratification- of course there were language/cultural reasons to do this but what other differences were expected. They mention Pakistani, Bangladeshi, Black in the background but not the other groups.

Response: We have aligned to C1 provided a justification in the methods section regarding the stratification of the focus groups, which reads ‘A total of 24 single sex focus groups were conducted based on self-identified ethnicity, with a combined sample of 110 parents (63 mothers and 47 fathers).  Stratifying focus groups by age, gender and ethnicity was particularly important to facilitate and enhance engagement with participants within the diverse and hard to reach communities [44, 45] whereby cultural and language barriers or gender segregation practices may have impacted on open participation [43, 46-48]’. We have also made clearer in the introduction where barriers reflect the white as well as Pakistani, Bangladeshi and African-Caribbean population.  A core focus of this study was to explore barriers/facilitators from the perspective of a culturally diverse and deprived community, stratification enabled us to determine where experiences were unique alongside where commonalities exist.

R1 C3: Second, more information regarding the facilitators, their training/credentials, and participant knowledge of the interviewer would add more information to the reader about personal characteristics of the research team and reflexibility.

Response: We have provided more information regarding the training/credentials of the facilitators: ‘All facilitators were recruited by the research team from the University of Bedfordshire and were experienced qualitative fieldworkers with a postgraduate qualification in health sciences.  Training events were delivered by the research team which provided an overview of the research study, presented the research instruments to be used and provided the facilitators an opportunity to practice the topic guide with trained researchers’.   None of the interviewers knew any of the participants ‘It is important to note that participants did not know facilitators prior to participation and focus groups were not attended by anyone outside of the facilitator and participants’.

R1 C4: Third, it was unclear what methodological orientation was utilized to underpin the study (grounded theory, discourse analysis, etc?) and if the principle of saturation was used as well. Explicitly stating these processes would provide the reader with more insights related to the study design.

Response: Thank you for raising this, we should have made this clearer this has now been added ‘This was a qualitative study with a phenomenological perspective using focus group discussions to generate in-depth contextualised information from a range of opinions and experiences’ and this has also been added to the abstract.  We should also note that the principle of saturation was used which has now also been added: ‘The number of focus groups held was driven by saturation, where consecutive focus groups revealed no additional second-level categories [39]

R1 C5: Fourth, while I understand that the study is very specific to UK and these ethnic groups- there is a broader literature with regards to immigrants, change in eating, food, parenting etc. that could be included in the discussion to provide additional comparison/context.

Response: We would like to agree with the point raised and personally thank the reviewer for highlighting relevant evidence which can further develop our discussions and link to broader international evidence.  We have now added the aforementioned studies and provided comparison in the discussion where relevant. 

R1 C6: We recommend that the authors consider referencing the COREQ checklist as this may add rigor to their process reporting: http://cdn.elsevier.com/promis_misc/ISSM_COREQ_Checklist.pdf.

Response: We have reviewed the COREQ and have used this to guide the reporting of this article.

R1 C7: Abstract: the first sentence (lines 16-17) could be broken up into two sentences to increase clarity for the reader.

Response: This sentence has now been broken into two sentences and now reads ‘In the UK ethnic minority children are at greater risk of obesity and weight-related ill health compared to the wider national population.  The factors that influence the provision of a healthy diet among these populations remain less understood.’

R1 C8: Line 140: was this guide pilot tested? If so, need to delineate the process of pilot-testing.

The interview guide was pilot tested by the facilitators – this has now been added ‘Facilitators were asked to pilot the interview guide to firstly test the appropriateness of the questions but also to gain experience in using the guide to build rapport and facilitate discussions.  No modifications were made following this process’. 

R1 C9: Lines 155-157: this sentence is a little unclear. It seems like the focus groups were facilitated in English, although the facilitators were proficient in Punjabi, Pahari and Urdu; would break the sentences or reword for clarification

Response: This should have been clearer.  The facilitators were bilingual although participant’s chose mostly to speak in English although used some explanations in the vernacular.  This has now been revised ‘Focus groups with Pakistani and Bangladeshi mothers were facilitated by bilingual facilitators proficient in Punjabi, Pahari and Urdu.  However, participants mostly used English with some explanations in the vernacular’. 

R1 C10: Line 170-171: how were the audio recordings confirmed as being reflective of the interview? Did the facilitators give confirmation or were the researchers?

We asked all facilitators to check both the audio recording and the transcripts to provide confirmation that they were reflective of the focus group they conducted.  This has now been added: All facilitators were asked to check the audio recordings to confirm they were reflective of the focus groups they conducted, with all transcripts given to the participants for comment’.

R1 C11: Please provide justification for same sex groups

This has now been added ‘Same sex focus groups were particularly important as we were engaging with diverse and hard to reach communities [41, 42] which may hold gender segregation practices which may have impacted on open participation [43, 44]’. 

R1 C12: Discussion: line 585 – unclear if it’s a title?

Response: This has now been deleted.

Reviewer 2 Report

The manuscript presents a very important theme, providing information that is extremely useful in terms of food security and in the design of food policies and other interventions.

Line 80 - The authors stated that there is also some evidence indicating that there is poor knowledge of healthy diet and weight among some BAME groups. Although this is data supported by the literature, it is important to reflect on why, namely the different impact of interventions to promote healthy eating. It would also be an important introduction point for the work itself.

Was the study approved by any ethics committee? the information should be available

Table 2 - The code "Practical skills and strategies" was placed in the Time and financial constraints topic instead of Knowledge, skills, and capabilities. What is the reason?

Discussion/ Conclusion: Bearing in mind that the data were collected between 2014 and 2015 (almost 8 years ago) and are only published now, a current reflection in terms of the evolution of the determinants presented would be very important. We know that contexts and environments are constantly changing, taking into account the covid-19 pandemic that certainly had consequences for environmental determinants.

Author Response

R2 C1: Line 80 - The authors stated that there is also some evidence indicating that there is poor knowledge of healthy diet and weight among some BAME groups. Although this is data supported by the literature, it is important to reflect on why, namely the different impact of interventions to promote healthy eating. It would also be an important introduction point for the work itself.

Response: We have acknowledged the role of food perceptions/beliefs and how this may link with poor knowledge of healthy diet and weight among some BAME groups.  This now reads ‘There is also some evidence suggesting that there is poor knowledge of healthy diet and weight among some BAME groups [27], particularly around awareness of traditional foods which have high levels of saturated fat and sugar [28].  Food attitudes and beliefs are also shown to shape dietary behaviours among some ethnic minority populations with perceptions that traditional diets are healthier than Western diets [28-30]’.    

R2 C2: Was the study approved by any ethics committee? the information should be available

Response: This study was approved by the University of Bedfordshire Institute for Health Research ethics committee, and we have provided the Institutional Review Board Statement: ‘This study was conducted according to the guidelines of the Declaration of Helsinki and all procedures involving research study participants were approved by the University of Bedfordshire Institute for Health Research ethics committee (IHRREC367)’. We have also added the sentence ‘Ethical approval was provided by the University of Bedfordshire Institute for Health Research ethics committee (IHRREC367)’ in the data methods section. 

R2 C3: Table 2 - The code "Practical skills and strategies" was placed in the Time and financial constraints topic instead of Knowledge, skills, and capabilities. What is the reason?

Response: We have now moved this sub code to the ‘Knowledge, skills, and capabilities’.

R2 C4: Discussion/ Conclusion: Bearing in mind that the data were collected between 2014 and 2015 (almost 8 years ago) and are only published now, a current reflection in terms of the evolution of the determinants presented would be very important. We know that contexts and environments are constantly changing, considering the covid-19 pandemic that certainly had consequences for environmental determinants.

Response: This is an important and valid point and so we have acknowledged this in our concluding paragraph ‘It is also important to acknowledge that the contexts and environments are constantly changing, particularly given the recent covid-19 pandemic which would have had additional consequences for environmental determinants’.

Reviewer 3 Report

I really enjoyed the study titled "‘They are kids, let them eat’: A qualitative investigation into the parental beliefs and practices of providing a healthy diet for young children among a culturally diverse and deprived population in the UK ". This is an interesting study and very well written by the authors. I have some suggestions and comments that can be incorporated to improve the clarity and scientific explanation of this paper.   

Abstract:

Please explain what is (aged 0-5)? years or months?

Introduction:

The introduction is very well written and organized according to the research objective. 

Material and methods:

There are several qualitative research designs such as: grounded theory, ethnography, phenomenology etc. Mention any relevant qualitative research design instead of just mentioning "used a qualitative interpretative research design using focus group discussions". Moreover, qualitative research is a research paradigm instead of a design. So, choose a relevant qualitative research design for this study. 

Did the researcher not interview Indians? Indians may be there apart from Pakistani and Bangladeshi populations.

Table 01 depicts that the authors mentioned only 18 FGDs. However, in the abstract and sampling section, the authors mentioned that they collected 24 FGD.

Moreover, table 01 can explain the brief socio-demographic characteristics of study participants as this manuscript explains. https://pubmed.ncbi.nlm.nih.gov/33866982/

However, following the above pattern is not strongly recommended. 

Furthermore, I would suggest summarizing table 01 according to ethnic groups instead of explaining each FGD. For example, Pakistani, Bangladeshi, Black African, and polish. Insert another column and just mention how many FGDs were collected from each ethnic group.

Data collection

Just mention the interview guide instead of the interview topic guide.

Figure 01 can be mentioned as supplementary material in Microsoft word format.

Line #146-148 should be mentioned at the start of the method section with proper citation.

Data analysis

Can the authors explain a bit about the framework approach? 

Table 02 also mentions where authors are referring to in data analysis?

Moreover, how data were analyzed? Either deductive method or indicative method?

Results: 

Remain consistent with the format. For example, L 225-228, the participant's identifier is missing.

Authors can use figures or any framework to better present their result findings instead of table 02. However, this is not strongly recommended. 

The result, discussion, and conclusions are well written. 

Author Response

R3 C1: Please explain what is (aged 0-5)? years or months?

Response: This has now been clarified ‘0-5 years’

R3 C2: There are several qualitative research designs such as: grounded theory, ethnography, phenomenology etc. Mention any relevant qualitative research design instead of just mentioning "used a qualitative interpretative research design using focus group discussions". Moreover, qualitative research is a research paradigm instead of a design. So, choose a relevant qualitative research design for this study. 

Response: Thank you for raising this, we should have made this clearer this has now been added ‘This was a qualitative study with a phenomenological perspective using focus group discussions to generate in-depth contextualised information from a range of opinions and experiences’ and this has also been added to the abstract.

R3 C3: Did the researcher not interview Indians? Indians may be there apart from Pakistani and Bangladeshi populations.

Response: Whilst we appreciate that Indians represent a significant Asian population within the UK; this study focused only on Pakistani and Bangladeshi populations as these groups were found more likely to reside in the most deprived communities in Luton. 

R3 C4: Table 01 depicts that the authors mentioned only 18 FGDs. However, in the abstract and sampling section, the authors mentioned that they collected 24 FGD.

Response: Thank you for identifying this.  There were 24 focus groups, but we omitted some of the Polish focus groups in the Table.  We have now amended this, and Table 01 is now accurate.

R3 C5: Moreover, table 01 can explain the brief socio-demographic characteristics of study participants as this manuscript explains. https://pubmed.ncbi.nlm.nih.gov/33866982/ However, following the above pattern is not strongly recommended. Furthermore, I would suggest summarizing table 01 according to ethnic groups instead of explaining each FGD. For example, Pakistani, Bangladeshi, Black African, and polish. Insert another column and just mention how many FGDs were collected from each ethnic group.

Response: We have organised Table 01 around the five ethnic groups as advised and then provided information regarding gender/age composition and number of participants for each FG.

R3 C6: Just mention the interview guide instead of the interview topic guide.

Response: We have removed ‘topic’ so this now reads interview guide.

R3 C7: Figure 01 can be mentioned as supplementary material in Microsoft word format.

Response: We have now removed Figure 01 and replaced as a supplementary file which is now in word format.

R3 C8: Line #146-148 should be mentioned at the start of the method section with proper citation.

Response: This sentence has now been moved and citations are added outlining the previous studies published:

Cook EJ, Powell FC, Ali N, Penn-Jones C, Ochieng B, Randhawa G: Parents’ experiences of complementary feeding among a United Kingdom culturally diverse and deprived community. Maternal & Child Nutrition 2020, 17:e13108.

Cook EJ, Powell F, Ali N, Penn-Jones C, Ochieng B, Randhawa G: Improving support for breastfeeding mothers: a qualitative study on the experiences of breastfeeding among mothers who reside in a deprived and culturally diverse community. International Journal for Equity in Health 2021, 20:1-14.

R3 C9: Can the authors explain a bit about the framework approach? 

Response: A short explanation of framework approach is now provided ‘Focus groups were analysed using the Framework Approach (FA) [46] which is a grounded and generative analytical procedure particularly useful in identifying commonalities and differences in qualitative data [47]’. 

R3 C10: Table 02 also mentions where authors are referring to in data analysis?

Response: We have moved Table 02, so it is presented with the data analysis section.

R3 C11: Moreover, how data were analyzed? Either deductive method or indicative method?

Response: Data was analysed inductively, and this has now been added ‘Coding was inductively applied manually to all narrative, which varied from small sections of data (parts of sentences) to whole paragraphs’.

R3 C12: Remain consistent with the format. For example, L 225-228, the participant's identifier is missing.

Response: This was a discussion between two fathers, so we added (Pakistani fathers, aged 31-45) at the end of this exchange.  We have added ‘South Asian fathers, in contrast as discussed below’ to make this clearer for the reader.

R3 C13: Authors can use figures or any framework to better present their result findings instead of table 02. However, this is not strongly recommended. 

Response: We would like to thank you for this suggestion.  Whilst figures can be a nice way to visually present findings, we have discussed this and are overall happy with the current presentation

Round 2

Reviewer 1 Report

Overall, I believe the authors adequately addressed the comments/concerns raised

during the first review of the manuscript. The paper currently provides a stronger

justification for stratification of participants/focus groups and reports their methodology

in a greater detail.

There were some minor comments/suggestions that would aid in the clarity of the paper

and the concepts discussed in it. These are detailed below:

Introduction:

Line 47-49: would consider rewording sentence for a better grammatical flow.

Lines 82-91: the first sentence on this portion (lines 82-85) feels contradictory to what

was mentioned earlier in this section. First, it was discussed that dietary acculturation is

associated with lower diet quality, and how children and second-generation immigrants

have lower diet quality than individuals who recently migrated or continue to reside in

their origin country. But then, this sentence makes it seem that traditional foods (not the

“Western” foods) are the “unhealthy” ones (higher levels of saturated fats and sugar),

which can perpetuate the misconception of having to get rid of traditional foods to be

“healthy”.

Lines 86-88: would consider rewording sentence for a better grammatical flow.

Line 104: would say “beliefs and perceptions”

Materials and methods:

Line 240: would replace “as a good will gesture” with “upon completion of their

participation/the study).

Conclusion:

Lines 735-737: mention of COVID-19 felt abrupt as this was not discussed earlier in the

paper, and I would consider removing it. However, if authors would like to keep it, I

would discuss this in the “discussion” section rather than in the “conclusion” section

given it would be pertinent to explain those environmental factors that have changed

during the pandemic (e.g., higher rates of food insecurity)

Author Response

We would firstly like to thank you for your useful feedback.  We have addressed all comments as set out below:

R1 C1: Line 47-49: would consider rewording sentence for a better grammatical flow.

Response: This has now been changed to ‘The increasing number of children becoming obese, particularly those from disadvantaged backgrounds will continue to represent an increasing economic burden both nationally and globally’. 

R1 C2: Lines 82-91: the first sentence on this portion (lines 82-85) feels contradictory to what

was mentioned earlier in this section. First, it was discussed that dietary acculturation is

associated with lower diet quality, and how children and second-generation immigrants

have lower diet quality than individuals who recently migrated or continue to reside in

their origin country. But then, this sentence makes it seem that traditional foods (not the

“Western” foods) are the “unhealthy” ones (higher levels of saturated fats and sugar),

which can perpetuate the misconception of having to get rid of traditional foods to be

“healthy”.

Response: We can this reads quite contradictory.  The point we were trying to make is that there are some traditional foods which may not be as healthy as that perceived so we have tried to make it clearer that not all traditional foods are unhealthy, although some may be.  We have also revised the next sentence so that it does not suggest that Westernised diets are healthier.  This section now reads: There is also some evidence suggesting that there is poor knowledge of healthy diet and weight among some BAME groups [29], particularly around awareness of traditional foods which may have high levels of saturated fat and sugar [30].  Parental dietary attitudes, beliefs and behaviour of migrated parents are also shown to shape those of their children [31] with a common perception that their ‘traditional’ diet is more healthier than the diet of their ‘host’ country [30, 32, 33]’.

R1 C3: Lines 86-88: would consider rewording sentence for a better grammatical flow.

Response: This has been changed to ‘Parental dietary attitudes, beliefs and behaviour of migrated parents are also shown to shape those of their children [31] with a common perception that their ‘traditional’ diet is more healthier than the diet of their ‘host’ country [30, 32, 33]’.

R1 C4: Line 104: would say “beliefs and perceptions”

Response: This has now been changed to ‘This study is a smaller sub study of a larger research programme which explored parents’ beliefs and perceptions on healthy diet and weight for children aged 0-5 years [36, 37].’

R1 C5: Line 240: would replace “as a good will gesture” with “upon completion of their

participation/the study).

Response: This has now been changed to ‘All participants were given a £10 high street voucher upon completion of their participation in the study.

R1 C6: Lines 735-737: mention of COVID-19 felt abrupt as this was not discussed earlier in the

paper, and I would consider removing it. However, if authors would like to keep it, I

would discuss this in the “discussion” section rather than in the “conclusion” section

given it would be pertinent to explain those environmental factors that have changed

during the pandemic (e.g., higher rates of food insecurity)

Response: This sentence has now been removed.